# TENSOR GRAPH CONVOLUTIONAL NETWORKS FOR PREDICTION ON DYNAMIC GRAPHS

## ABSTRACT

Many irregular domains such as social networks, financial transactions, neuron connections, and natural language structures are represented as graphs. In recent years, a variety of graph neural networks (GNNs) have been successfully applied for representation learning and prediction on such graphs. However, in many of the applications, the underlying graph changes over time and existing GNNs are inadequate for handling such dynamic graphs. In this paper we propose a novel technique for learning embeddings of dynamic graphs based on a tensor algebra framework. Our method extends the popular graph convolutional network (GCN) for learning representations of dynamic graphs using the recently proposed tensor M-product technique. Theoretical results that establish the connection between the proposed tensor approach and spectral convolution of tensors are developed. Numerical experiments on real datasets demonstrate the usefulness of the proposed method for an edge classification task on dynamic graphs.

## 1 INTRODUCTION

Graphs are popular data structures used to effectively represent interactions and structural relationships between entities in structured data domains. Inspired by the success of deep neural networks for learning representations in the image and language domains, recently, application of neural networks for graph representation learning has attracted much interest. A number of graph neural network (GNN) architectures have been explored in the contemporary literature for a variety of graph related tasks and applications (Hamilton et al., 2017; Seo et al., 2018; Chen et al., 2018; Zhou et al., 2018; Wu et al., 2019). Methods based on graph convolution filters which extend convolutional neural networks (CNNs) to irregular graph domains are popular (Bruna et al., 2013; Defferrard et al., 2016; Kipf and Welling, 2016). Most of these GNN models operate on a given, static graph.

In many real-world applications, the underlining graph changes over time, and learning representations of such dynamic graphs is essential. Examples include analyzing social networks (Berger-Wolf and Saia, 2006), predicting collaboration in citation networks (Leskovec et al., 2005), detecting fraud and crime in financial networks (Weber et al., 2018; Pareja et al., 2019), traffic control (Zhao et al., 2019), and understanding neuronal activities in the brain (De Vico Fallani et al., 2014). In such dynamic settings, the temporal interdependence in the graph connections and features also play a substantial role. However, efficient GNN methods that handle time varying graphs and that capture the temporal correlations are lacking.

By *dynamic graph*, we mean a sequence of graphs $(V, \mathbf{A}^{(t)}, \mathbf{X}^{(t)})$, $t \in \{1, 2, \ldots, T\}$, with a fixed set $V$ of $N$ nodes, adjacency matrices $\mathbf{A}^{(t)} \in \mathbb{R}^{N \times N}$, and graph feature matrices $\mathbf{X}^{(t)} \in \mathbb{R}^{N \times F}$ where $\mathbf{X}_{n:}^{(t)} \in \mathbb{R}^F$ is the feature vector consisting of $F$ features associated with node $n$ at time $t$. The graphs can be weighted, and directed or undirected. They can also have additional properties like (time varying) node and edge classes, which would be stored in a separate structure. Suppose we only observe the first $T' < T$ graphs in the sequence. The goal of our method is to use these observations to predict some property of the remaining $T - T'$ graphs. In this paper, we use it for edge classification. Other potential applications are node classification and edge/link prediction.

In recent years, tensor constructs have been explored to effectively process high-dimensional data, in order to better leverage the multidimensional structure of such data (Kolda and Bader, 2009). Tensor based approaches have been shown to perform well in many image and video processing ap-

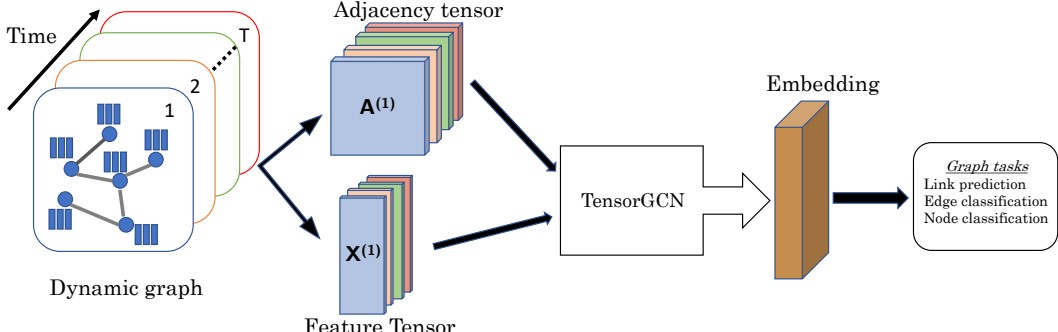

Figure 1: TensorGCN approach.

plications (Hao et al., 2013; Kilmer et al., 2013; Martin et al., 2013; Zhang et al., 2014; Zhang and Aeron, 2016; Lu et al., 2016; Newman et al., 2018). A number of tensor based neural networks have also been investigated to extract and learn multi-dimensional representations, e.g. methods based on tensor decomposition (Phan and Cichocki, 2010), tensor-trains (Novikov et al., 2015; Stoudenmire and Schwab, 2016), and tensor factorized neural network (Chien and Bao, 2017). Recently, a new tensor framework called the *tensor M-product framework* (Braman, 2010; Kilmer and Martin, 2011; Kernfeld et al., 2015) was proposed that extends matrix based theory to high-dimensional architectures.

In this paper, we propose a novel tensor variant of the popular graph convolutional network (GCN) architecture (Kipf and Welling, 2016), which we call *TensorGCN*. It captures correlation over time by leveraging the tensor M-product framework. The flexibility and matrix mimeticability of the framework, help us adapt the GCN architecture to tensor space. Figure 1 illustrates our method at a high level: First, the time varying adjacency matrices $\mathbf{A}^{(t)}$ and feature matrices $\mathbf{X}^{(t)}$ of the dynamic graph are aggregated into an adjacency tensor and a feature tensor, respectively. These tensors are then fed into our TensorGCN, which computes an embedding that can be used for a variety of tasks, such as link prediction, and edge and node classification. GCN architectures are motivated by graph convolution filtering, i.e., applying filters/functions to the graph Laplacian (in turn its eigenvalues) (Bruna et al., 2013), and we establish a similar connection between TensorGCN and spectral filtering of tensors. Experimental results on real datasets illustrate the performance of our method for the edge classification task on dynamic graphs. Elements of our method can also be used as a preprocessing step for other dynamic graph methods.

## 2    RELATED WORK

The idea of using graph convolution based on the spectral graph theory for GNNs was first introduced by Bruna et al. (2013). Defferrard et al. (2016) then proposed *Chebnet*, where the spectral filter was approximated by Chebyshev polynomials in order to make it faster and localized. Kipf and Welling (2016) presented the simplified GCN, a degree-one polynomial approximation of Chebnet, in order to speed up computation further and improve the performance. There are many other works that deal with GNNs when the graph and features are fixed/static; see the review papers by Zhou et al. (2018) and Wu et al. (2019) and references therein. These methods cannot be directly applied to the dynamic setting we consider. Seo et al. (2018) devised the Graph Convolutional Recurrent Network for graphs with time varying features. However, this method assumes that the edges are fixed over time, and is not applicable in our setting. Wang et al. (2018) proposed a method called EdgeConv, which is a neural network (NN) approach that applies convolution operations on static graphs in a dynamic fashion. Their approach is not applicable when the graph itself is dynamic. Zhao et al. (2019) develop a temporal GCN method called T-GCN, which they apply for traffic prediction. Their method assumes the graph remains fixed over time, and only the features vary.

The set of methods most relevant to our setting of learning embeddings of dynamic graphs use combinations of GNNs and recurrent architectures (RNN), to capture the graph structure and handle time dynamics, respectively. The approach in Manessi et al. (2019) uses Long Short-Term Memory (LSTM), a recurrent network, in order to handle time variations along with GNNs. They design

architectures for semi-supervised node classification and for supervised graph classification. Pareja et al. (2019) presented a variant of GCN called *EvolveGCN*, where Gated Recurrent Units (GRU) and LSTMs are coupled with a GCN to handle dynamic graphs. This paper is currently the state-of-the-art. However, their approach is based on a heuristic RNN/GRU mechanism, which is not theoretically viable, and does not harness a tensor algebraic framework to incorporate time varying information. Newman et al. (2018) present a tensor NN which utilizes the M-product tensor framework. Their approach can be applied to image and other high-dimensional data that lie on regular grids, and differs from ours since we consider data on dynamic graphs.

## 3 TENSOR M-PRODUCT FRAMEWORK

Here, we cover the necessary preliminaries on tensors and the M-product framework. For a more general introduction to tensors, we refer the reader to the review paper by Kolda and Bader (2009). In this paper, a *tensor* is a three-dimensional array of real numbers denoted by boldface Euler script letters, e.g. $\mathcal{X} \in \mathbb{R}^{I \times J \times T}$. Matrices are denoted by bold uppercase letters, e.g. $\mathbf{X}$; vectors are denoted by bold lowercase letter, e.g. $\mathbf{x}$; and scalars are denoted by lowercase letters, e.g. $x$. An element at position $(i, j, t)$ in a tensor is denotes by subscripts, e.g. $\mathcal{X}_{ijt}$, with similar notation for elements of matrices and vectors. A colon will denote all elements along that dimension; $\mathbf{X}_{i:}$ denotes the $i$th row of the matrix $\mathbf{X}$, and $\mathcal{X}_{::k}$ denotes the $k$th frontal slice of $\mathcal{X}$. The vectors $\mathcal{X}_{ij:}$ are called the *tubes* of $\mathcal{X}$.

The framework we consider relies on a new definition of the product of two tensors, called the M-product (Braman, 2010; Kilmer and Martin, 2011; Kilmer et al., 2013; Kernfeld et al., 2015). A distinguishing feature of this framework is that the M-product of two three-dimensional tensors is also three-dimensional, which is not the case for e.g. tensor contractions (Bishop and Goldberg, 2012). It allows one to elegantly generalize many classical numerical methods from linear algebra, and has been applied e.g. in neural networks (Newman et al., 2018), imaging (Kilmer et al., 2013; Martin et al., 2013; Semerci et al., 2014), facial recognition (Hao et al., 2013), and tensor completion and denoising (Zhang et al., 2014; Zhang and Aeron, 2016; Lu et al., 2016). Although the framework was originally developed for three-dimensional tensors, which is sufficient for our purposes, it has been extended to handle tensors of dimension greater than three (Martin et al., 2013). The following definitions 3.1–3.3 describe the M-product.

**Definition 3.1** (M-transform). Let $\mathbf{M} \in \mathbb{R}^{T \times T}$ be a mixing matrix. The *M-transform* of a tensor $\mathcal{X} \in \mathbb{R}^{I \times J \times T}$ is denoted by $\mathcal{X} \times_3 \mathbf{M} \in \mathbb{R}^{I \times J \times T}$ and defined elementwise as

$$(\mathcal{X} \times_3 \mathbf{M})_{ijt} \stackrel{\text{def}}{=} \sum_{k=1}^{T} \mathbf{M}_{tk} \mathcal{X}_{ijk}. \tag{1}$$

We say that $\mathcal{X} \times_3 \mathbf{M}$ is in the *transformed space*. Note that if $\mathbf{M}$ is invertible, then $(\mathcal{X} \times_3 \mathbf{M}) \times_3 \mathbf{M}^{-1} = \mathcal{X}$. Consequently, $\mathcal{X} \times_3 \mathbf{M}^{-1}$ is the *inverse M-transform* of $\mathcal{X}$. The definition in (1) may also be written in matrix form as $\mathcal{X} \times_3 \mathbf{M} \stackrel{\text{def}}{=} \text{fold}(\mathbf{M} \, \text{unfold}(\mathcal{X}))$, where the unfold operation takes the tubes of $\mathcal{X}$ and stack them as columns into a $T \times IJ$ matrix, and $\text{fold}(\text{unfold}(\mathcal{X})) = \mathcal{X}$. Appendix A provides illustrations of how the M-transform works.

**Definition 3.2** (Facewise product). Let $\mathcal{X} \in \mathbb{R}^{I \times J \times T}$ and $\mathcal{Y} \in \mathbb{R}^{J \times K \times T}$ be two tensors. The *facewise product*, denote by $\mathcal{X} \triangle \mathcal{Y} \in \mathbb{R}^{I \times K \times T}$, is defined facewise as $(\mathcal{X} \triangle \mathcal{Y})_{::t} \stackrel{\text{def}}{=} \mathcal{X}_{::t} \mathcal{Y}_{::t}$.

**Definition 3.3** (M-product). Let $\mathcal{X} \in \mathbb{R}^{I \times J \times T}$ and $\mathcal{Y} \in \mathbb{R}^{J \times K \times T}$ be two tensors, and let $\mathbf{M} \in \mathbb{R}^{T \times T}$ be an invertible matrix. The *M-product*, denoted by $\mathcal{X} \star \mathcal{Y} \in \mathbb{R}^{I \times K \times T}$, is defined as

$$\mathcal{X} \star \mathcal{Y} \stackrel{\text{def}}{=} ((\mathcal{X} \times_3 \mathbf{M}) \triangle (\mathcal{Y} \times_3 \mathbf{M})) \times_3 \mathbf{M}^{-1}.$$

In the original formulation of the M-product, $\mathbf{M}$ was chosen to be the Discrete Fourier Transform (DFT) matrix, which allows efficient computation using the Fast Fourier Transform (FFT) (Braman, 2010; Kilmer and Martin, 2011; Kilmer et al., 2013). The framework was later extended for arbitrary invertible $\mathbf{M}$ (e.g. discrete cosine and wavelet transforms) (Kernfeld et al., 2015). A benefit of the tensor M-product framework is that many standard matrix concepts can be generalized in a straightforward manner. Definitions 3.4–3.7 extend the matrix concepts of diagonality, identity, transpose and orthogonality to tensors (Braman, 2010; Kilmer et al., 2013).

**Definition 3.4** (f-diagonal). A tensor $\mathcal{X} \in \mathbb{R}^{N \times N \times T}$ is said to be *f-diagonal* if each frontal slice $\mathcal{X}_{::t}$ is diagonal.

**Definition 3.5** (Identity tensor). Let $\hat{\mathcal{J}} \in \mathbb{R}^{N \times N \times T}$ be defined facewise as $\hat{\mathcal{J}}_{::t} = \mathbf{I}$, where $\mathbf{I}$ is the matrix identity. The M-product *identity tensor* $\mathcal{J} \in \mathbb{R}^{N \times N \times T}$ is then defined as $\mathcal{J} \stackrel{\text{def}}{=} \hat{\mathcal{J}} \times_3 \mathbf{M}^{-1}$.

**Definition 3.6** (Tensor transpose). The transpose of a tensor $\mathcal{X}$ is defined as $\mathcal{X}^\top \stackrel{\text{def}}{=} \mathcal{Y} \times_3 \mathbf{M}^{-1}$, where $\mathcal{Y}_{::t} = (\mathcal{X} \times_3 \mathbf{M})_{::t}^\top$ for each $t \in \{1, \dots, T\}$.

**Definition 3.7** (Orthogonal tensor). A tensor $\mathcal{X} \in \mathbb{R}^{N \times N \times T}$ is said to be *orthogonal* if $\mathcal{X} \star \mathcal{X}^\top = \mathcal{X}^\top \star \mathcal{X} = \mathcal{J}$.

Leveraging these concepts, a tensor eigendecomposition can now be defined (Braman, 2010; Kilmer et al., 2013):

**Definition 3.8** (Tensor eigendecomposition). Let $\mathcal{X} \in \mathbb{R}^{N \times N \times T}$ be a tensor and assume that each frontal slice $(\mathcal{X} \times_3 \mathbf{M})_{::t}$ is symmetric. We can then eigendecompose these as $(\mathcal{X} \times_3 \mathbf{M})_{::t} = \hat{\mathcal{Q}}_{::t} \hat{\mathcal{D}}_{::t} \hat{\mathcal{Q}}_{::t}^\top$, where $\hat{\mathcal{Q}}_{::t} \in \mathbb{R}^{N \times N}$ is orthogonal and $\hat{\mathcal{D}}_{::t} \in \mathbb{R}^{N \times N}$ is diagonal (see e.g. Theorem 8.1.1 in Golub and Van Loan (2013)). The *tensor eigendecomposition* of $\mathcal{X}$ is then defined as $\mathcal{X} \stackrel{\text{def}}{=} \mathcal{Q} \star \mathcal{D} \star \mathcal{Q}^\top$, where $\mathcal{Q} \stackrel{\text{def}}{=} \hat{\mathcal{Q}} \times_3 \mathbf{M}^{-1}$ is orthogonal, and $\mathcal{D} \stackrel{\text{def}}{=} \hat{\mathcal{D}} \times_3 \mathbf{M}^{-1}$ if f-diagonal.

## 4 TENSOR DYNAMIC GRAPH EMBEDDING

Our approach is inspired by the first order GCN by Kipf and Welling (2016) for static graphs, owed to its simplicity and effectiveness. For a graph with adjacency matrix $\mathbf{A}$ and feature matrix $\mathbf{X}$, a GCN layer takes the form $\mathbf{Y} = \sigma(\tilde{\mathbf{A}} \mathbf{X} \mathbf{W})$, where

$$\tilde{\mathbf{A}} \stackrel{\text{def}}{=} \tilde{\mathbf{D}}^{-1/2}(\mathbf{A} + \mathbf{I})\tilde{\mathbf{D}}^{-1/2},$$

$\tilde{\mathbf{D}}$ is diagonal with $\tilde{\mathbf{D}}_{ii} = 1 + \sum_j \mathbf{A}_{ij}$, $\mathbf{I}$ is the matrix identity, $\mathbf{W}$ is a matrix to be learned when training the NN, and $\sigma$ is an activation function, e.g., ReLU. Our approach translates this to a tensor model by utilizing the M-product framework. We first introduce a tensor activation function $\hat{\sigma}$ which operates in the transformed space.

**Definition 4.1.** Let $\mathcal{A} \in \mathbb{R}^{I \times J \times T}$ be a tensor and $\sigma$ an elementwise activation function. We define the activation function $\hat{\sigma}$ as $\hat{\sigma}(\mathcal{A}) \stackrel{\text{def}}{=} \sigma(\mathcal{A} \times_3 \mathbf{M}) \times_3 \mathbf{M}^{-1}$.

We can now define our proposed dynamic graph embedding. Let $\mathcal{A} \in \mathbb{R}^{N \times N \times T}$ be a tensor with frontal slices $\mathcal{A}_{::t} = \tilde{\mathbf{A}}^{(t)}$, where $\tilde{\mathbf{A}}^{(t)}$ is the normalization of $\mathbf{A}^{(t)}$. Moreover, let $\mathcal{X} \in \mathbb{R}^{N \times F \times T}$ be a tensor with frontal slices $\mathcal{X}_{::t} = \mathbf{X}^{(t)}$. Finally, let $\mathcal{W} \in \mathbb{R}^{F \times F' \times T}$ be a weight tensor. We define our dynamic graph embedding as $\mathcal{Y} = \mathcal{A} \star \mathcal{X} \star \mathcal{W} \in \mathbb{R}^{N \times F' \times T}$. This computation can also be repeated in multiple layers. For example, a 2-layer formulation would be of the form

$$\mathcal{Y} = \mathcal{A} \star \hat{\sigma}(\mathcal{A} \star \mathcal{X} \star \mathcal{W}^{(0)}) \star \mathcal{W}^{(1)}.$$

One important consideration is how to choose the matrix $\mathbf{M}$ which defines the M-product. For time-varying graphs, we choose $\mathbf{M}$ to be lower triangular and banded so that each frontal slice $(\mathcal{A} \times_3 \mathbf{M})_{::t}$ is a linear combination of the adjacency matrices $\mathcal{A}_{::\max(1,t-b+1)}, \dots, \mathcal{A}_{::t}$, where we refer to $b$ as the "bandwidth" of $\mathbf{M}$. This choice ensures that each frontal slice $(\mathcal{A} \times_3 \mathbf{M})_{::t}$ only contains information from current and past graphs that are close temporally. Specifically, the entries of $\mathbf{M}$ are set to

$$\mathbf{M}_{tk} \stackrel{\text{def}}{=} \begin{cases} \frac{1}{\min(b,t)} & \text{if } \max(1, t-b+1) \le k \le t, \\ 0 & \text{otherwise,} \end{cases}$$

which implies that $\sum_k \mathbf{M}_{tk} = 1$ for each $t$. Another possibility is to treat $\mathbf{M}$ as a parameter matrix to be learned from the data.

In order to avoid over-parameterization and improve the performance, we choose the weight tensor $\mathcal{W}$ (at each layer), such that each of the frontal slices of $\mathcal{W}$ in the transformed domain remains the same, i.e., $(\mathcal{W} \times_3 \mathbf{M})_{::t} = (\mathcal{W} \times_3 \mathbf{M})_{::t'} \; \forall t, t'$. In other words, the parameters in each layer

are shared and learned over all the training instances. This reduces the number of parameters to be learned significantly.

An embedding $\mathcal{Y} \in \mathbb{R}^{N \times F' \times T}$ can now be used for various prediction tasks, like link prediction, and edge and node classification. In Section 5, we apply our method for edge classification by using a model similar to that used by Pareja et al. (2019): Given an edge between nodes $m$ and $n$ at time $t$, the predictive model is

$$p(m, n, t) \stackrel{\text{def}}{=} \text{softmax}(\mathbf{U}[(\mathcal{Y} \times_3 \mathbf{M})_{m:t}, (\mathcal{Y} \times_3 \mathbf{M})_{n:t}]^\top),$$

where $(\mathcal{Y} \times_3 \mathbf{M})_{m:t} \in \mathbb{R}^{F'}$ and $(\mathcal{Y} \times_3 \mathbf{M})_{n:t} \in \mathbb{R}^{F'}$ are row vectors, $\mathbf{U} \in \mathbb{R}^{C \times 2F'}$ is a weight matrix, and $C$ the number of classes. Note that the embedding $\mathcal{Y}$ is first M-transformed before the matrix $\mathbf{U}$ is applied to the appropriate feature vectors. This, combined with the fact that the tensor activation functions are applied elementwise in the transformed domain, allow us to avoid ever needing to apply the inverse M-transform. This approach reduces the computational cost, and has been found to improve performance in the edge classification task.

## 4.1 THEORETICAL MOTIVATION FOR TENSORGCN

Here, we present the results that establish the connection between the proposed TensorGCN and spectral convolution of tensors, in particular spectral filtering and approximation on dynamic graphs. This is analogous to the graph convolution based on spectral graph theory in the GNNs by Bruna et al. (2013), Defferrard et al. (2016), and Kipf and Welling (2016). All proofs are provided in Appendix D.

Let $\mathcal{L} \in \mathbb{R}^{N \times N \times T}$ be a form of tensor Laplacian defined as $\mathcal{L} \stackrel{\text{def}}{=} \mathcal{I} - \mathcal{A}$. Throughout the remainder of this subsection, we will assume that the adjacency matrices $\mathbf{A}^{(t)}$ are symmetric.

**Proposition 4.2.** *The tensor $\mathcal{L}$ has an eigendecomposition $\mathcal{L} = \mathcal{Q} \star \mathcal{D} \star \mathcal{Q}^\top$.*

Much like the spectrum of a normalized graph Laplacian is contained in $[0, 2]$ (Shuman et al., 2013), the tensor spectrum of $\mathcal{L}$ satisfies a similar property.

**Proposition 4.3** (Spectral bound). *The entries of $\hat{\mathcal{D}} = \mathcal{D} \times_3 \mathbf{M}$ lie in $[0, 2]$.*

Following the work by Kilmer et al. (2013), three-dimensional tensors in $\mathbb{R}^{M \times N \times T}$ can be viewed as operators on $N \times T$ matrices, with those matrices "twisted" into tensors in $\mathbb{R}^{N \times 1 \times T}$. With this in mind, we define a tensor variant of the graph Fourier transform.

**Definition 4.4** (Tensor-tube M-product). Let $\mathcal{X} \in \mathbb{R}^{I \times J \times T}$ and $\boldsymbol{\theta} \in \mathbb{R}^{1 \times 1 \times T}$. Analogously to the definition of the matrix-scalar product, we define $\mathcal{X} \star \boldsymbol{\theta}$ via $(\mathcal{X} \star \boldsymbol{\theta})_{ij:} \stackrel{\text{def}}{=} \mathcal{X}_{ij:} \star \boldsymbol{\theta}$.

**Definition 4.5** (Tensor graph Fourier transform). Let $\mathcal{X} \in \mathbb{R}^{N \times F \times T}$ be a tensor. We define a *tensor graph Fourier transform $F$* as $F(\mathcal{X}) \stackrel{\text{def}}{=} \mathcal{Q}^\top \star \mathcal{X} \in \mathbb{R}^{N \times F \times T}$.

This is analogous to the definition of the matrix graph Fourier transform. This defines a convolution like operation for tensors similar to spectral graph convolution (Shuman et al., 2013; Bruna et al., 2013). Each lateral slice $\mathcal{X}_{:j:}$ is expressible in terms of the set $\{\mathcal{Q}_{:n:}\}_{n=1}^N$ as follows:

$$\mathcal{X}_{:j:} = \mathcal{Q} \star \mathcal{Q}^\top \star \mathcal{X}_{:j:} = \sum_{n=1}^N \mathcal{Q}_{:n:} \star (\mathcal{Q}^\top \star \mathcal{X}_{:j:})_{n1:},$$

where each $(\mathcal{Q}^\top \star \mathcal{X}_{:j:})_{n1:} \in \mathbb{R}^{1 \times 1 \times T}$ can be considered a tubal scalar. In fact, the lateral slices $\mathcal{Q}_{:n:}$ form a basis for the set $\mathbb{R}^{N \times 1 \times T}$ with product $\star$; see Appendix D for further details.

**Definition 4.6** (Tensor spectral graph filtering). Given a signal $\mathcal{X} \in \mathbb{R}^{N \times 1 \times T}$ and a function $g : \mathbb{R}^{1 \times 1 \times T} \to \mathbb{R}^{1 \times 1 \times T}$, we define the *tensor spectral graph filtering* of $\mathcal{X}$ with respect to $g$ as

$$\mathcal{X}_{\text{filt}} \stackrel{\text{def}}{=} \mathcal{Q} \star g(\mathcal{D}) \star \mathcal{Q}^\top \star \mathcal{X}, \tag{2}$$

where

$$g(\mathcal{D})_{mn:} \stackrel{\text{def}}{=} \begin{cases} g(\mathcal{D}_{mn:}) & \text{if } m = n, \\ \mathbf{0} & \text{if } m \neq n. \end{cases}$$

In order to avoid the computation of an eigendecomposition, Defferrard et al. (2016) use a polynomial to approximate the filter function. We take a similar approach, and approximate $g(\mathcal{D})$ with an M-product polynomial. For this approximation to make sense, we impose additional structure on $g$.

**Assumption 4.7.** Assume that $g : \mathbb{R}^{1 \times 1 \times T} \to \mathbb{R}^{1 \times 1 \times T}$ is defined as

$$g(\mathcal{V}) \stackrel{\text{def}}{=} f(\mathcal{V} \times_3 \mathbf{M}) \times_3 \mathbf{M}^{-1},$$

where $f$ is defined elementwise as $f(\mathcal{V} \times_3 \mathbf{M})_{11t} \stackrel{\text{def}}{=} f^{(t)}((\mathcal{V} \times_3 \mathbf{M})_{11t})$ with each $f^{(t)} : \mathbb{R} \to \mathbb{R}$ continuous.

**Proposition 4.8.** *Suppose $g$ satisfies Assumption 4.7. For any $\varepsilon > 0$, there exists an integer $K$ and a set $\{\boldsymbol{\theta}^{(k)}\}_{k=1}^{K} \subset \mathbb{R}^{1 \times 1 \times T}$ such that*

$$\left\| g(\mathcal{D}) - \sum_{k=0}^{K} \mathcal{D}^{\star k} \star \boldsymbol{\theta}^{(k)} \right\| < \varepsilon,$$

*where $\| \cdot \|$ is the tensor Frobenius norm, and where $\mathcal{D}^{\star k} \stackrel{\text{def}}{=} \mathcal{D} \star \cdots \star \mathcal{D}$ is the M-product of $k$ instances of $\mathcal{D}$, with the convention that $\mathcal{D}^{\star 0} = \mathcal{I}$.*

As in the work of Defferrard et al. (2016), a tensor polynomial approximation allows us to approximate $\mathcal{X}_{\text{filt}}$ in (2) without computing the eigendecomposition of $\mathcal{L}$:

$$\mathcal{X}_{\text{filt}} = \mathcal{Q} \star g(\mathcal{D}) \star \mathcal{Q}^{\top} \star \mathcal{X} \approx \mathcal{Q} \star \left( \sum_{k=0}^{K} \mathcal{D}^{\star k} \star \boldsymbol{\theta}^{(k)} \right) \star \mathcal{Q}^{\top} \star \mathcal{X} = \left( \sum_{k=0}^{K} \mathcal{L}^{\star k} \star \boldsymbol{\theta}^{(k)} \right) \star \mathcal{X}. \quad (3)$$

All that is necessary is to compute tensor powers of $\mathcal{L}$. We can also define tensor polynomial analogs of the Chebyshev polynomials and do the approximation in (3) in terms of those instead of the tensor monomials $\mathcal{D}^{\star k}$. This is not necessary for the purposes of this paper. Instead, we note that if a degree-one approximation is used, the computation in (3) becomes

$$\mathcal{X}_{\text{filt}} \approx (\mathcal{I} \star \boldsymbol{\theta}^{(0)} + \mathcal{L} \star \boldsymbol{\theta}^{(1)}) \star \mathcal{X} = (\mathcal{I} \star \boldsymbol{\theta}^{(0)} + (\mathcal{I} - \mathcal{A}) \star \boldsymbol{\theta}^{(1)}) \star \mathcal{X}.$$

Setting $\boldsymbol{\theta} \stackrel{\text{def}}{=} \boldsymbol{\theta}^{(0)} = -\boldsymbol{\theta}^{(1)}$, which is analogous to the parameter choice made in the degree-one approximation by Kipf and Welling (2016), we get

$$\mathcal{X}_{\text{filt}} \approx \mathcal{A} \star \mathcal{X} \star \boldsymbol{\theta}. \quad (4)$$

If we let $\mathcal{X}$ contain $F$ signals, i.e., $\mathcal{X} \in \mathbb{R}^{N \times F \times T}$, and apply $F'$ filters, (4) becomes

$$\mathcal{X}_{\text{filt}} \approx \mathcal{A} \star \mathcal{X} \star \boldsymbol{\Theta} \in \mathbb{R}^{N \times F' \times T},$$

where $\boldsymbol{\Theta} \in \mathbb{R}^{F \times F' \times T}$. This is precisely our embedding model, with $\boldsymbol{\Theta}$ replaced by a learnable parameter tensor $\mathcal{W}$.

## 5 NUMERICAL EXPERIMENTS

Here, we present results for edge classification on four datasets[1]: The Bitcoin Alpha and OTC transaction datasets (Kumar et al., 2016), the Reddit body hyperlink dataset (Kumar et al., 2018), and a chess results dataset (Kunegis, 2013). The bitcoin datasets consist of transaction histories for users on two different platforms. Each node is a user, and each directed edge indicates a transaction and is labeled with an integer between $-10$ and $10$ which indicates the senders trust for the receiver. We convert these labels to two classes: positive (trustworthy) and negative (untrustworthy). The Reddit dataset is build from hyperlinks from one subreddit to another. Each node represents a subreddit, and each directed edge is an interaction which is labeled with $-1$ for a hostile interaction or $+1$ for a friendly interaction. We only consider those subreddits which have a total of 20 interactions or more. In the chess dataset, each node is a player, and each directed edge represents a match with the source node being the white player and the target node being the black player. Each edge is labeled $-1$ for a black victory, $0$ for a draw, and $+1$ for a white victory. Table 1 summarizes the statistics for the different datasets.

---

[1]We provide links to the datasets in Appendix B.

Table 1: Dataset statistics.

| Dataset | Nodes | Edges | Graphs ($T$) | Time window length | Classes |
|---|---|---|---|---|---|
| Bitcoin OTC | 6,005 | 35,569 | 135 | 14 days | 2 |
| Bitcoin Alpha | 7,604 | 24,173 | 135 | 14 days | 2 |
| Reddit | 3,818 | 163,008 | 86 | 14 days | 2 |
| Chess | 7,301 | 64,958 | 100 | 31 days | 3 |

The data is temporally partitioned into $T$ graphs, with each graph containing data from a particular time window. Both $T$ and the time window length can vary between datasets. For each node-time pair $(n, t)$ in these graphs, we compute the number of outgoing and incoming edges and use these two numbers as features. The adjacency tensor $\mathcal{A}$ is then constructed as described in Section 4. The $T$ frontal slices of $\mathcal{A}$ are divided into $S_{\text{train}}$ training slices, $S_{\text{val}}$ validation slices, and $S_{\text{test}}$ testing slices, which come sequentially after each other; see Figure 2 and Table 2.

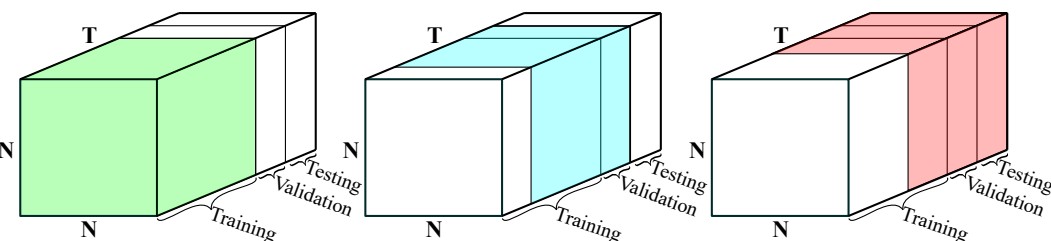

Figure 2: Partitioning of $\mathcal{A}$ into training, validation and testing data.

Table 2: Partitioning and performance metric for each dataset.

| Dataset | Partitioning | | | Performance metric |
|---|---|---|---|---|
| | $S_{\text{train}}$ | $S_{\text{val}}$ | $S_{\text{test}}$ | |
| Bitcoin OTC | 95 | 20 | 20 | F1 score |
| Bitcoin Alpha | 95 | 20 | 20 | F1 score |
| Reddit | 66 | 10 | 10 | F1 score |
| Chess | 80 | 10 | 10 | Accuracy |

Since the adjacency matrices corresponding to graphs are very sparse for these datasets, we apply the same technique as Pareja et al. (2019) and add the entries of each frontal slice $\mathcal{A}_{::t}$ to the following $l-1$ frontal slices $\mathcal{A}_{::t}, \ldots, \mathcal{A}_{::(t+l-1)}$, where we refer to $l$ as the "edge life." Note that this only affects $\mathcal{A}$, and that the added edges are not treated as real edges in the classification problem.

The bitcoin and Reddit datasets are heavily skewed, with about 90% of edges labeled positively, and the remaining labeled negatively. Since the negative instances are more interesting to identify (e.g. to prevent financial fraud or online hostility), we use the F1 score to evaluate the experiments on these datasets, treating the negative edges as the ones we want to identify. The classes are more well-balanced in the chess dataset, so we use accuracy to evaluate those experiments.

We choose to use an embedding $\mathcal{Y}_{\text{train}} = \mathcal{A}_{::(1:S_{\text{train}})} \star \mathcal{X}_{::(1:S_{\text{train}})} \star \mathcal{W}$ for training. When computing the embeddings for the validation and testing data, we still need $S_{\text{train}}$ frontal slices of $\mathcal{A}$, which we get by using a sliding window of slices. This is illustrated in Figure 2, where the green, blue and red blocks show the frontal slices used when computing the embeddings for the training, validation and testing data, respectively. The embeddings for the validation and testing data are $\mathcal{Y}_{\text{val}} = \mathcal{A}_{::(S_{\text{val}}+1:S_{\text{train}}+S_{\text{val}})} \star \mathcal{X}_{::(S_{\text{val}}+1:S_{\text{train}}+S_{\text{val}})} \star \mathcal{W}$ and $\mathcal{Y}_{\text{test}} = \mathcal{A}_{::(S_{\text{val}}+S_{\text{test}}+1:T)} \star \mathcal{X}_{::(S_{\text{val}}+S_{\text{test}}+1:T)} \star \mathcal{W}$, respectively. Preliminary experiments with 2-layer architectures did not show convincing improvements in performance. We believe this is due to the fact that the datasets only have two features, and that a 1-layer architecture therefore is sufficient for extracting relevant information in the data. For training, we use the cross entropy loss function:

$$\text{loss} = -\sum_t \sum_{(m,n) \in E_t} \sum_{c=1}^{C} \alpha_c f(m, n, t)_c \log(p(m, n, t)_c), \tag{5}$$

Table 3: Results without symmetrizing adjacency matrices. A higher value is better.

| Method | Dataset | | | |
|---|---|---|---|---|
| | Bitcoin OTC | Bitcoin Alpha | Reddit | Chess |
| WD-GCN | 0.2062 | 0.1920 | **0.2337** | 0.4311 |
| EvolveGCN | 0.3284 | 0.1609 | 0.2012 | 0.4351 |
| GCN | 0.3317 | 0.2100 | 0.1805 | 0.4342 |
| TensorGCN (Proposal) | **0.3529** | **0.2331** | 0.2028 | **0.4708** |

Table 4: Results when using symmetrized adjacency matrices. A higher value is better.

| Method | Dataset | | | |
|---|---|---|---|---|
| | Bitcoin OTC | Bitcoin Alpha | Reddit | Chess |
| WD-GCN | 0.1009 | 0.1319 | **0.2173** | 0.4321 |
| EvolveGCN | 0.0913 | **0.2273** | 0.1942 | 0.4091 |
| GCN | 0.0769 | 0.1538 | 0.1966 | 0.4369 |
| TensorGCN (Proposal) | **0.3103** | 0.2207 | 0.2071 | **0.4713** |

where $f(m, n, t) \in \mathbb{R}^C$ is a one-hot vector encoding the true class of the edge $(m, n)$ at time $t$, and $\alpha \in \mathbb{R}^C$ is a vector summing to 1 which contains the weight of each class. Since the bitcoin and Reddit datasets are so skewed, we weigh the minority class more heavily in the loss function for those datasets, and treat $\alpha$ as a hyperparameter; see Appendix C for details.

The experiments are implemented in PyTorch with some preprocessing done in Matlab. Our code is available at [url redacted for review]. In the experiments, we use an edge life of $l = 10$, a bandwidth $b = 20$, and $F' = 6$ output features. Since the graphs in the considered datasets are directed, we also investigate the impact of symmetrizing the adjacency matrices, where the symmetrized version of an adjacency matrix $\mathbf{A}$ is defined as $\mathbf{A}_{\mathrm{sym}} \overset{\mathrm{def}}{=} 1/2(\mathbf{A} + \mathbf{A}^\top)$.

We compare our method with three other methods. The first one is a variant of the WD-GCN by Manessi et al. (2019), which they specify in Equation (8a) of their paper. For the LSTM layer in their description, we use 6 output features instead of $N$. This is to avoid overfitting and make the method more comparable to ours which uses 6 output features. For the final layer, we use the same prediction model as that used by Pareja et al. (2019) for edge classification. The second method is a 1-layer variant of EvolveGCN-H by Pareja et al. (2019). The third method is a simple baseline which uses a 1-layer version of the GCN by Kipf and Welling (2016). It uses the same weight matrix $\mathbf{W}$ for all temporal graphs. Both EvolveGCN-H and the baseline GCN use 6 output features as well.

Table 3 shows the results when the adjacency matrices have not been symmetrized. In this case, our method outperforms the other methods on the two bitcoin datasets and the chess dataset, with WD-GCN performing best on the Reddit dataset. Table 4 shows the results for when the adjacency matrices have been symmetrized. Our method outperforms the other methods on the Bitcoin OTC dataset and the chess dataset, and performs similarly but slightly worse than the best performing methods on the Bitcoin Alpha and Reddit datasets. Overall, it seems like symmetrizing the adjacency matrices leads to lower performance.

## 6    CONCLUSION

We have presented a novel approach for dynamic graph embedding which leverages the tensor M-product framework. We used it for edge classification in experiments on four real datasets, where it performed competitively compared to state-of-the-art methods. Future research directions include further developing the theoretical guarantees for the method, investigating optimal structure and learning of the transform matrix $\mathbf{M}$, using the method for other prediction tasks, and investigating how to utilize deeper architectures for dynamic graph learning.

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

# Appendices

## A  ILLUSTRATION OF THE M-TRANSFORM

We provide some illustrations that show how the M-transform in Definition 3.1 works. Recall that $\mathcal{X} \times_3 \mathbf{M} = \mathrm{fold}(\mathbf{M}\,\mathrm{unfold}(\mathcal{X}))$. The matrix $\mathcal{X}$ is first unfolded into a matrix, as illustrated in Figure 3. This unfolded tensor is then multiplied from the left by the matrix $\mathbf{M}$, as illustrated in Figure 4; the figure also illustrates the banded lower triangular structure of $\mathbf{M}$. Finally, the output matrix is folded back into a tensor. The fold operation is defined to be the inverse of the unfold operation.

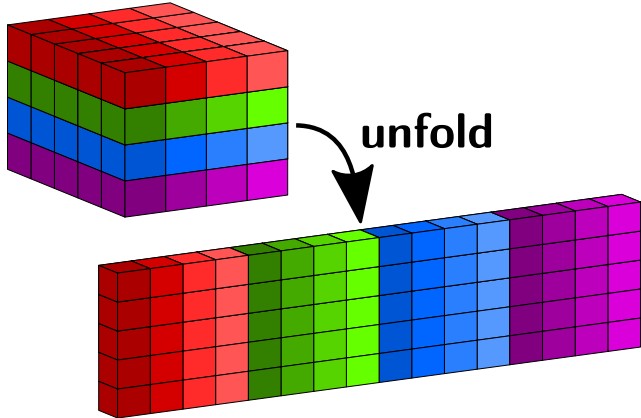

Figure 3: Illustration of unfold operation applied to $4 \times 4 \times 5$ tensor.

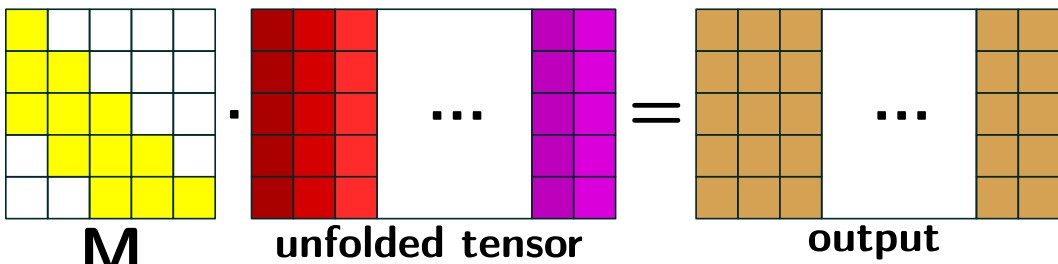

Figure 4: Illustration of matrix product between $\mathbf{M}$ and the unfolded tensor.

## B  LINKS TO DATASETS

- The Bitcoin Alpha dataset is available at
  https://snap.stanford.edu/data/soc-sign-bitcoin-alpha.html.

- The Bitcoin OTC dataset is available at
  https://snap.stanford.edu/data/soc-sign-bitcoin-otc.html.

- The Reddit dataset is available at
  https://snap.stanford.edu/data/soc-RedditHyperlinks.html.
  Note that we use the dataset with hyperlinks in the body of the posts.

- The chess dataset is available at
  http://konect.uni-koblenz.de/networks/chess.

## C    FURTHER DETAILS ON THE EXPERIMENT SETUP

When partitioning the data into $T$ graphs, as described in Section 5, if there are multiple data points corresponding to an edge $(m, n)$ for a given time step $t$, we only add that edge once to the corresponding graph and set the label equal to the sum of the labels of the different data points. For example, if bitcoin user $m$ makes three transactions to $n$ during time step $t$ with ratings 10, 2, $-1$, then we add a single edge $(m, n)$ to graph $t$ with label $10 + 2 - 1 = 11$.

For training, we run gradient descent with a learning rate of 0.01 and momentum of 0.9 for 10,000 iterations. For each 100 iterations, we compute and store the performance of the model on the validation data. As mentioned in Section 5, the weight vector $\alpha$ in the loss function (5) is treated as a hyperparameter in the bitcoin and Reddit experiments. Since these datasets all have two edge classes, let $\alpha_0$ and $\alpha_1$ be the weights of the minority (negative) and majority (positive) classes, respectively. Since these parameters add to 1, we have $\alpha_1 = 1 - \alpha_0$. For all methods, we repeat the bitcoin and Reddit experiments once for each $\alpha_0 \in \{0.75, 0.76, \ldots, 0.95\}$. For each model and dataset, we then find the best stored performance of the model on the validation data across all $\alpha_0$ values. We then treat the corresponding model as the trained model, and report its performance on the testing data in Tables 3 and 4. The results for the chess experiment are computed in the same way, but only for a single vector $\alpha = [1/3,\ 1/3,\ 1/3]$.

## D    ADDITIONAL RESULTS AND PROOFS

Throughout this section, $\| \cdot \|$ will denote the Frobenius norm (i.e., the square root of the sum of the elements squared) of a matrix or tensor, and $\| \cdot \|_2$ will denote the matrix spectral norm.

We first provide a few further results that clarify the algebraic properties of the M-product. Let $\mathbb{R}^{1 \times 1 \times T}$ denote the set of $1 \times 1 \times T$ tensors. Similarly, let $\mathbb{R}^{N \times 1 \times T}$ denote the set of $N \times 1 \times T$ tensors. Under the M-product framework, the set $\mathbb{R}^{1 \times 1 \times T}$ play a role similar to that played by scalars in matrix algebra. With this in mind, the set $\mathbb{R}^{N \times 1 \times T}$ can be seen as a length $N$ vector consisting of tubal elements of length $T$. Propositions D.1 and D.2 make this more precise.

**Proposition D.1** (Proposition 4.2 in Kernfeld et al. (2015))**.** *The set $\mathbb{R}^{1 \times 1 \times T}$ with product $\star$, which is denoted by $(\star, \mathbb{R}^{1 \times 1 \times T})$, is a commutative ring with identity.*

**Proposition D.2** (Theorem 4.1 in Kernfeld et al. (2015))**.** *The set $\mathbb{R}^{N \times 1 \times T}$ with product $\star$, which is denoted by $(\star, \mathbb{R}^{N \times 1 \times T})$, is a free module over the ring $(\star, \mathbb{R}^{1 \times 1 \times T})$.*

A free module is similar to a vector space. Like a vector space, it has a basis. Proposition D.3 shows that the lateral slices of $\mathcal{Q}$ in the tensor eigendecomposition form a basis for $(\star, \mathbb{R}^{N \times 1 \times T})$, similarly to how the eigenvectors in a matrix eigendecomposition form a basis.

**Proposition D.3.** *The lateral slices $\mathcal{Q}_{:n:} \in \mathbb{R}^{N \times 1 \times T}$ of $\mathcal{Q}$ in Definition 3.8 form a basis for $(\star, \mathbb{R}^{N \times 1 \times T})$.*

*Proof.* Let $\mathcal{X} \in \mathbb{R}^{N \times 1 \times T}$. Note that

$$\mathcal{X} = \mathcal{I} \star \mathcal{X} = \mathcal{Q} \star \mathcal{Q}^\top \star \mathcal{X} = \sum_{n=1}^{N} \mathcal{Q}_{:n:} \star \mathcal{V}_{n1:},$$

where $\mathcal{V} \stackrel{\text{def}}{=} \mathcal{Q}^\top \star \mathcal{X} \in \mathbb{R}^{N \times 1 \times T}$. So the lateral slices of $\mathcal{Q}$ are a generating set for $(\star, \mathbb{R}^{N \times 1 \times T})$. Now suppose

$$\sum_{n=1}^{N} \mathcal{Q}_{:n:} \star \mathcal{S}_{n1:} = \mathbf{0},$$

for some $\mathcal{S} \in \mathbb{R}^{N \times 1 \times T}$. Then $\mathbf{0} = \mathcal{Q} \star \mathcal{S}$, and consequently

$$\mathbf{0} = (\mathcal{Q} \times_3 \mathbf{M}) \triangle (\mathcal{S} \times_3 \mathbf{M}).$$

Since each frontal face of $\mathcal{Q} \times_3 \mathbf{M}$ is an invertible matrix, this implies that each frontal face of $\mathcal{S} \times_3 \mathbf{M}$ is zero, and hence $\mathcal{S} = \mathbf{0}$. So the lateral slices of $\mathcal{Q}$ are also linearly independent in $(\star, \mathbb{R}^{N \times 1 \times T})$. □

### D.1 PROOFS OF PROPOSITIONS IN THE MAIN TEXT

*Proof of Proposition 4.2.* Since each adjacency matrix $\mathbf{A}^{(t)}$ and each $\mathcal{I}_{::t}$ is symmetric, each frontal slice $\mathcal{L}_{::t}$ is also symmetric. Consequently,

$$(\mathcal{L} \times_3 \mathbf{M})_{ij:} = \mathcal{L}_{ij:} \times_3 \mathbf{M} = \mathcal{L}_{ji:} \times_3 \mathbf{M} = (\mathcal{L} \times_3 \mathbf{M})_{ji:},$$

so each frontal slice of $\mathcal{L} \times_3 \mathbf{M}$ is symmetric, and therefore $\mathcal{L}$ has an eigendecomposition. □

*Proof of Proposition 4.3.* Each $\mathcal{A}_{::t}$ has a spectrum contained in $[-1, 1]$. Since $\mathcal{A}_{::t}$ is symmetric, it follows that $\|\mathcal{A}_{::t}\|_2 \leq 1$. Consequently,

$$\|(\mathcal{A} \times_3 \mathbf{M})_{::t}\|_2 = \left\| \sum_{j=1}^{T} \mathbf{M}_{tj} \mathcal{A}_{::j} \right\|_2 \leq \sum_{j=1}^{T} |\mathbf{M}_{tj}| \|\mathcal{A}_{::j}\|_2 \leq 1,$$

where we used the fact that $\sum_j |\mathbf{M}_{tj}| = 1$. So since the frontal slices $(\mathcal{A} \times_3 \mathbf{M})_{::t}$ are symmetric, they each have a spectrum in $[-1, 1]$. It follows that each frontal slice

$$(\mathcal{L} \times_3 \mathbf{M})_{::t} = \mathbf{I} - (\mathcal{A} \times_3 \mathbf{M})_{::t}$$

has a spectrum contained in $[0, 2]$, which means that the entries of $\hat{\mathcal{D}}$ all lie in $[0, 2]$. □

**Lemma D.4.** *Let $\mathcal{X} \in \mathbb{R}^{M \times N \times T}$ and let $\mathbf{M} \in \mathbb{R}^{T \times T}$ be invertible. Then*

$$\|\mathcal{X}\| \leq \|\mathbf{M}^{-1}\|_2 \|\mathcal{X} \times_3 \mathbf{M}\|.$$

*Proof.* We have

$$\|\mathcal{X}\| = \|(\mathcal{X} \times_3 \mathbf{M}) \times_3 \mathbf{M}^{-1}\| = \|\mathbf{M}^{-1} \operatorname{unfold}(\mathcal{X} \times_3 \mathbf{M})\|$$
$$\leq \|\mathbf{M}^{-1}\|_2 \| \operatorname{unfold}(\mathcal{X} \times_3 \mathbf{M})\| = \|\mathbf{M}^{-1}\|_2 \|\mathcal{X} \times_3 \mathbf{M}\|,$$

where the inequality is a well-known relation that holds for all matrices. □

*Proof of Proposition 4.8.* By Weierstrass approximation theorem, there exists an integer $K$ and a set $\{\hat{\boldsymbol{\theta}}^{(k)}\}_{k=1}^{K} \subset \mathbb{R}^{1 \times 1 \times T}$ such that for all $t \in \{1, 2, \ldots, T\}$,

$$\sup_{x \in [0,2]} \left| f^{(t)}(x) - \sum_{k=0}^{K} x^k \hat{\boldsymbol{\theta}}_{11t}^{(k)} \right| < \frac{\varepsilon}{\|\mathbf{M}^{-1}\|_2 \sqrt{NT}}.$$

Let $\boldsymbol{\theta}^{(k)} \stackrel{\text{def}}{=} \hat{\boldsymbol{\theta}}^{(k)} \times_3 \mathbf{M}^{-1}$. Note that if $m \neq n$, then

$$\left( \sum_{k=0}^{K} \mathcal{D}^{\star k} \star \boldsymbol{\theta}^{(k)} \right)_{mn:} = \sum_{k=0}^{K} ((\hat{\mathcal{D}}^{\triangle k})_{mn:} \times_3 \mathbf{M}^{-1}) \star \boldsymbol{\theta}^{(k)} = \mathbf{0} = g(\mathcal{D})_{mn:},$$

since $\hat{\mathcal{D}} = \mathcal{D} \times_3 \mathbf{M}$ is f-diagonal. So

$$\left\| g(\mathcal{D}) - \sum_{k=0}^{K} \mathcal{D}^{\star k} \star \boldsymbol{\theta}^{(k)} \right\|^2 = \sum_{n=1}^{N} \left\| g(\mathcal{D})_{nn:} - \sum_{k=0}^{K} (\mathcal{D}^{\star k})_{nn:} \star \boldsymbol{\theta}^{(k)} \right\|^2$$
$$\leq \|\mathbf{M}^{-1}\|_2^2 \sum_{n=1}^{N} \left\| g(\mathcal{D})_{nn:} \times_3 \mathbf{M} - \sum_{k=0}^{K} ((\mathcal{D} \times_3 \mathbf{M})^{\triangle k})_{nn:} \triangle \hat{\boldsymbol{\theta}}^{(k)} \right\|^2$$
$$= \|\mathbf{M}^{-1}\|_2^2 \sum_{n=1}^{N} \sum_{t=1}^{T} \left| f^{(t)}((\mathcal{D} \times_3 \mathbf{M})_{nnt}) - \sum_{k=0}^{K} (\mathcal{D} \times_3 \mathbf{M})_{nnt}^k \hat{\boldsymbol{\theta}}_{11t}^{(k)} \right|^2$$
$$< \varepsilon^2,$$

where the first inequality follows from Lemma D.4. Taking square roots completes the proof. □

