# OpenReview forum: "Tensor Graph Convolutional Networks for Prediction on Dynamic Graphs"
_ICLR.cc/2020/Conference — Reject_

### Official Review · AnonReviewer3 · 2019-10-24
**Official Blind Review #3**

**Rating:** 6

**Review:**

Summary: this work uses tensor methods to improve graph convolution for dynamic graph, where the nodes are fixed and the edges are changing. Specifically, it uses the M-product technique to develop the operations of sequence of matrices that analog to these operations of matrices. In the M-product notations, everything seems to be as neat as matrix operations. The works also shows decent supremacy on edge classification tasks.


Comments: this paper is mathematically interesting. It is well-written in general, but the definitions are dense and hard to follow.

It will be better to give some examples of M-product. For example, what these operations will be if we choose M to be the identity matrix?

M-transfer is a tensor contraction, right?

It seems if you do the operations of the sequence of matrix, there is no need to do iterations like RNN. I am interested in how this will influence the runtime and memory cost.

The M matrix is defined as a lower triangle matrix such as (A \times M)_::t depends on A^(1:t). Is it possible to formulate M such that (A \times M)_::t will depend heavily on A^(t), and less on the farther matices? such that we encode some Markov property?

Does there exist some condition when this method will be equivalent to RNN?


Decision: I feel this work novel and interesting in general. I would like to weakly accept it.


**Experience Assessment:**

I have read many papers in this area.

**Review Assessment: Checking Correctness Of Derivations And Theory:**

I assessed the sensibility of the derivations and theory.

**Review Assessment: Checking Correctness Of Experiments:**

I assessed the sensibility of the experiments.

**Review Assessment: Thoroughness In Paper Reading:**

I read the paper thoroughly.

---

> ### Author Response · Authors · 2019-11-08
> **Response to review #3**
>
> Thank you for reading our paper and providing feedback.
>
> - Giving concrete examples of M is a good idea. We will include further examples of concrete choices of M (such as M = identity) and discuss what these mean. This discussion will be added to the appendix.
>
> - The M-transform wouldn't typically be called a tensor contraction since it does not completely collapse any dimensions or change the size of any dimension of the three-dimensional tensor (since M is square). The computation done in the M-transform is typically called a mode-3 tensor-times-matrix (TTM) product in the tensor literature.
>
> - Our hope is that our tensor approach will capture time dynamics and therefore eliminate the need for other time modeling like RNNs. However, you could also use elements (e.g. the M-transform) of our method as a preprocessing step before applying models incorporating RNN to it. This may increase performance, and is an area for future exploration. Depending on how M is chosen, the cost could potentially be reduced compared a model which incorporates an RNN. This all depends on how the different models are parameterized.
>
> - It is indeed possible to encode some notion of a Markov property by varying the bandwidth and weight of the matrix M. For example, the magnitude of the elements in each row of M could decrease exponentially as we move to the left of the main diagonal. This would, in a sense, encode that data further in the past is much less important than more current data.
>
> - This is a great question. It seems more likely that they are not equivalent, except for possibly in a very trivial case. However, there may be ways to adapt our method by adding more components that make them more similar and even equivalent. This is an interesting direction for future research.

---

### Official Review · AnonReviewer1 · 2019-10-24
**Official Blind Review #1**

**Rating:** 3

**Review:**

This paper presents a M-product based temporal GCNs to handle dynamic graphs. Experiments on four real datasets are performed to verify the effectiveness of the proposed model.

Overall, I think this paper make a few contributions to advocate tensor M-product. However, there are several big issues as listed below. Given the current status, I could not accept the paper.

Pros:

1, The generalization brought by M-product seems to be general as it includes quite a few graph convolution elements for 3D tensors in a natural way.

2, The experimental setup is reasonable. Datasets are collected from practical problems and of moderately large scale.

3, The paper is clearly written and easy to follow.

Cons & Questions:

1, My first concern is that M-product formulation does not bring any new insights as people have already used some of the key elements in practice for a long time. For example, the M-transform is just applying 1x1 convolution to multi-channel image. Slice-wise matrix multiplication is also common in practice.

2, Moreover, I think there are several challenges in the M-product formulation which prevent the technique from being practical.

(1) Sharing M such that frontal slices of the transformed signal are the same, i.e., each row of M share the same vector, limits the model capacity significantly. If there is no sharing mechanism, then the model learned on one sequence of graphs could not be applied to another sequence of graphs given two sequences have different lengths.

(2) If you learn M from data, how could you ensure that M is invertible? In the paragraph before section 4.1, an edge classification formulation is proposed where the inverse M-transform is abandoned. However, if in practice, you do not need the inverse transform, then do those theoretical properties still hold and what is the meaning of introducing such M-product formulation?

3, A few temporal GCN baselines are neither compared or discussed, e.g., [1].

4, Could you explain why all the other GCN variants performs significantly worse with a symmetrized adjacency matrix compared to using the asymmetric one?

[1] Li, Y., Yu, R., Shahabi, C. and Liu, Y., 2017. Diffusion convolutional recurrent neural network: Data-driven traffic forecasting. arXiv preprint arXiv:1707.01926.

======================================================================================================

After I read authors' reply and other reviewers' comments, I would like to keep my original rating as the issues have not been properly addressed. I agree with the Reviewer #4 that the theoretical results are a bit artificial and trivial.

**Experience Assessment:**

I have published in this field for several years.

**Review Assessment: Checking Correctness Of Derivations And Theory:**

I carefully checked the derivations and theory.

**Review Assessment: Checking Correctness Of Experiments:**

I assessed the sensibility of the experiments.

**Review Assessment: Thoroughness In Paper Reading:**

I read the paper at least twice and used my best judgement in assessing the paper.

---

> ### Author Response · Authors · 2019-11-08
> **Response to review #1**
>
> Thank you for reading our paper and providing feedback.
>
> - Different elements of our method have indeed been used before. However, to the best of our knowledge, they have not been used together in the way that we do. Moreover, the framework we use brings together these various ideas into a principled approach with a sound theoretical foundation.
>
> - The only limitation on M is that it is invertible; the rows don't have to be the same. Indeed, our proposed M has rows that are different; see Fig. 4 in our paper.
>
> - If we learn M from data, we can impose various constraints to ensure that M is invertible (e.g., that M is diagonally dominant). We chose the specific form of the model for p(m,n,t) since it roughly corresponds to temporal mixing of the separate adjacency graphs and then applying a standard GCN to each of the new mixed adjacency graphs. We thought this simplicity was appealing as it makes the model more interpretable.
>
> - The paper [Li et al., 2017] considers a setting in which both the nodes and edges remain fixed over time. In our paper, the edges change over time. So the method by [Li et al., 2017] is not applicable, which is why we don't compare to it.
>
> - Intuitively, it seems natural that performance decreases as the adjacency matrices are symmetrized, since this destroys information about directionality on the graph. It could be that direction of a relationship is more important in the bitcoin datasets, which are more negatively impacted by symmetrization, than the Reddit and chess datasets.
>
> [Li et al., 2017] Li, Y., Yu, R., Shahabi, C. and Liu, Y., 2017. Diffusion convolutional recurrent neural network: Data-driven traffic forecasting. arXiv preprint arXiv:1707.01926.

---

### Official Review · AnonReviewer4 · 2019-10-30
**Official Blind Review #4**

**Rating:** 1

**Review:**

The paper proposed a new type of graph embedding technique for dynamic graphs based on tensor representation (node x feature x time).  Experiments on edge classification demonstrate improved prediction accuracy.

+ Clear writing with tensor notations and explanation is well-structured
+ Improved prediction results on 3/4 real-world dynamic graph datasets

- The theoretical results are a bit artificial. The tensor eigendecomposition used in this paper and [Kilmer and Martin] is for slices of the tensor, similarly for FFT and convolution. The technique is a trivial generalization from matrix results.
- The paper is missing a large body of baselines, both from the network science community (non-deep learning methods) and from this community (diffusion convolutional RNNs, graph attention networks, etc).
- The method doesn't scale well, especially for graphs with long-term dynamics. It would be good to show the scaling behavior of the proposed model.

**Experience Assessment:**

I have published in this field for several years.

**Review Assessment: Checking Correctness Of Derivations And Theory:**

I carefully checked the derivations and theory.

**Review Assessment: Checking Correctness Of Experiments:**

I carefully checked the experiments.

**Review Assessment: Thoroughness In Paper Reading:**

I read the paper thoroughly.

---

> ### Author Response · Authors · 2019-11-08
> **Response to review #4**
>
> Thank you for reading our paper and providing feedback.
>
> - The M-product has been developed to have matrix mimetic properties. The associated framework therefore provides a principled way to extend methods from the matrix setting to the tensor setting. This has previously been done for classical matrix computations like the SVD and QR decompositions. The fact that our conversion of the standard GCN to a time varying setting seems simple, or even trivial, reflects the power of the M-product framework that we utilize.
>
> - Aside from the simple GCN by [Kipf & Welling, 2016], we limit ourselves to comparing to methods that support time varying graphs (i.e., both edges and signals vary over time). We would be grateful if you could provide specific references to any such papers that we may have missed.
>
> - The main purpose of this paper is to propose a new model for handling time varying graphs, which few previous works have done. Due to the page limitation, we did not have the space to explore issues related to computational efficiency and scaling. However, we believe it is possible to make the method scale well with a careful choice of the mixing matrix M. Indeed, as a special case, if M is the identity, then our method is the same as the simple GCN by [Kipf & Welling, 2016] applied to each graph individually. By e.g. setting M to be to a fast transform (e.g., FFT, DCT, Wavelets) or maintaining sufficient sparsity (e.g., lower triangular with a narrow bandwidth), the method will scale almost as well as if M was the identity.
>
> [Kipf & Welling, 2016] Kipf, Thomas N., and Max Welling. "Semi-supervised classification with graph convolutional networks." arXiv preprint arXiv:1609.02907 (2016).

---

> > ### Comment · AnonReviewer4 · 2019-11-15
> > **Read the response**
> >
> > - It would be great if you can highlight the novel contributions on the tensor setting, other than repeating matrix computation multiple times.
> >
> > - The work assumes a fixed max graph, which is not so different from a fixed graph. I don't see why DCRNN/GAN cannot be modified to your setting.
> >
> > I read the response and would keep the review.

---

> > > ### Author Response · Authors · 2019-11-15
> > > **Follow up response**
> > >
> > > Our work assumes a fixed set of nodes, but time varying edges and node features. The DCRNN paper by [Li et al., 2017] assumes that both the nodes and edges remain fixed, which is a different setting. Can you please provide a specific reference to the GAN work you have in mind?
> > >
> > > [Li et al., 2017] Li, Y., Yu, R., Shahabi, C. and Liu, Y., 2017. Diffusion convolutional recurrent neural network: Data-driven traffic forecasting. arXiv preprint arXiv:1707.01926.

---

### Decision · Program_Chairs · 2019-12-19

**Decision:**

Reject

**Comment:**

The paper proposes a tensor-based extension to graph convolutional networks for prediction over dynamic graphs.

The proposed model is reasonable and achieves promising empirical results. After discussion, it is agreed that while the problem of handling dynamic graphs is interesting and challenging, the proposed tensor method lacks novelty, the theoretical analysis is artificial, and the empirical study does not cover enough benchmarks.

The current version of the paper is not ready for publication. Addressing the issues above could lead to a strong publication in the future.